# Sex Differences in E-Cigarette Use and Related Health Effects

**DOI:** 10.3390/ijerph20227079

**Published:** 2023-11-18

**Authors:** Fatima Alam, Patricia Silveyra

**Affiliations:** 1Department of Environmental and Occupational Health, Indiana University School of Public Health Bloomington, Bloomington, IN 47405, USA; alamfatima99@gmail.com; 2Department of Medicine, Indiana University School of Medicine, Indianapolis, IN 47405, USA

**Keywords:** e-cigarette, sex, gender, electronic cigarette, e-cigarette, vaping

## Abstract

Background: Electronic cigarettes (e-cigarettes) comprise a variety of products designed to deliver nicotine, flavorings, and other substances. To date, multiple epidemiological and experimental studies have reported a variety of health issues associated with their use, including respiratory toxicity, exacerbation of respiratory conditions, and behavioral and physiological effects. While some of these effects appear to be sex- and/or gender-related, only a portion of the research has been conducted considering these variables. In this review, we sought to summarize the available literature on sex-specific effects and sex and gender differences, including predictors and risk factors, effects on organ systems, and behavioral effects. Methods: We searched and selected articles from 2018–2023 that included sex as a variable or reported sex differences on e-cigarette-associated effects. Results: We found 115 relevant studies published since 2018 that reported sex differences in a variety of outcomes. The main differences reported were related to reasons for initiation, including smoking history, types of devices and flavoring, polysubstance use, physiological responses to nicotine and toxicants in e-liquids, exacerbation of lung disease, and behavioral factors such as anxiety, depression, sexuality, and bullying. Conclusions: The available literature supports the notion that both sex and gender influence the susceptibility to the negative effects of e-cigarette use. Future research needs to consider sex and gender variables when addressing e-cigarette toxicity and other health-related consequences.

## 1. Introduction

Electronic cigarettes (e-cigarettes) were introduced to the United States’ market in 2007. E-cigarettes comprise a variety of products that are constantly evolving. These currently come in the form of e-cigars, e-pipes, disposable e-cigarettes, rechargeable e-cigarettes, and large- and medium-sized tank devices [1]. The main components of these devices are solvents, flavorings, and nicotine. These e-cigarettes’ constituents play a role in affecting health, as indicated by multiple studies conducted in human populations, animal models, and cells [2]. In addition, the newest generation of e-cigarettes, called pod-devices, have pod cartridges filled with e-liquid and use nicotine salts that allow for higher concentrations of nicotine to be inhaled and delivered [3].

Vaping for smoking cessation, flavorings, experimentation, mood elevation, and weight loss are some of the reasons why e-cigarette use gained popularity among youth over the recent decade [1]. Despite mild recent declines in use among high school students, and slight increases in use in middle school students, e-cigarettes remain the most commonly used tobacco product in youth (7.7% of all students) [4]. The most recent survey indicates that among current e-cigarette users, 25% report using them daily with 89.4% choosing flavored products [4]. The current literature provides some evidence of successful cessation of cigarette smoking with e-cigarette use with factors such as flavor and degree of nicotine dependence positively impacting smoking cessation outcomes [5,6]. Also, a study on COPD smokers who switched to e-cigarettes from conventional cigarettes found a reduction in COPD exacerbations and an improvement in physical activity when compared to non-vaper COPD smokers [7]. Furthermore, a published randomized trial found that the e-cigarette group had better abstinence rates as well as better adherence to treatment compared to the nicotine replacement group [8]. Multiple studies have shown that the use of e-cigarettes containing high concentrations of nicotine has detrimental effects on various organ systems; however, the growing notion of e-cigarette use as a safer substitute to cigarette smoking has kept the lethal effects of vaping use guarded from the public [9]. Not only do these devices deliver a greater concentration of nicotine than conventional cigarettes, but also many currently available vape products contain tetrahydrocannabinol (THC) and other compounds that are unknown to the user [1]. In addition, most users, especially adolescents, have inadequate knowledge about e-cigarette-related health effects and the various dangerous chemicals that flavorings and other constituents produce.

To date, several detrimental health effects have been reported associated with e-cigarette use. These include effects on the respiratory, cardiovascular, and immune systems [9]. Importantly, effects of other environmental exposures on these organ systems have shown sex-specific effects, including females being more susceptible to cigarette smoke [10,11] and particulate air pollution exposure [12,13]. In this regard, sex has been recognized as an important modulator of inflammatory and immune responses to a variety of inhaled toxicants [14,15,16,17]. Despite these known disparities, whether there are sex-specific responses to e-cigarette exposures remains largely unknown.

In order to address this important gap in inhalation toxicology and summarize information that can educate researchers and users, we examined the available literature considering and incorporating the sex variable when determining risk factors, effects on organ systems, and other health effects of e-cigarette use. Our literature review not only identified studies reporting sex-specific biological effects, but also a number of studies addressing the gender variable in behavioral effects associated with e-cigarette use.

## 2. Materials and Methods

Database use and key terms searched: We performed a literature search on PubMed to identify eligible papers. We used the following search terms and logic: “sex” OR “sex differences” AND “electronic cigarette” OR “e-cigarette” OR “vaping”. All articles were recovered and selected based on the presence/absence of the search criteria.

Inclusion and exclusion criteria: The search was limited to papers written in English. We included publications from 2018 and later, and the final search was updated to September 2023. Articles were included if they tested sex-based differences. Articles were excluded if they discussed use of or effects of combustible cigarette smoking instead of e-cigarette smoking, if they addressed cigars, cigarillos, waterpipes, hookah, combustible cigs, mentholated cigarettes only, if they excluded e-cigarettes when conducting a study on polysubstance use, if they did not consider e-cigarettes as a separate variable or combined them with other smoking agents only when conducting experiments, and if there was a sex difference observed in the use of substances other than e-cigarettes.

Study selection and organization: All articles were recovered and selected based on the presence/absence of the search criteria. Articles were de-duplicated using the built-in mechanisms of university library services and further completed manually. Articles were assessed by their titles and abstracts for inclusion and organized in an Excel file. The final selections were determined after a complete reading of all articles and were organized using the Covidence systematic review software (Veritas Health Innovation, Melbourne, Australia).

Data extraction: The following information was extracted from the studies: the authors of the study, the year of publication, the study design, the sample size, and the main outcomes. Articles were then organized in tables according to specific outcomes related to e-cigarette exposures.

## 3. Results

### 3.1. Search Results

A summary of the search strategy and article selection is presented in Figure 1. The PubMed search identified 898 studies using the searched key terms. Of these, 506 studies were found to be duplicates. Of the remaining 392 studies, 258 were discarded as irrelevant due to an irrelevant study design, not including sex differences, or not addressing e-cigarette research. The full text of the remaining 134 articles was reviewed, and those that did not comply with the inclusion criteria were discarded. The final selection included 115 articles. These studies were subsequently divided into those studying sex differences in a) predictors and risk factors for e-cigarette use, b) effects on organ systems, and c) behavioral effects (including gender differences) (Table 1). In the latter category, several studies used the terms sex (biological variable) and gender (social construct) interchangeably, without including a definition of these terms. Therefore, articles using both terms were included. Participants of studies incorporating the gender variable identified as either male/female, bisexual male/female, gay/lesbian, transgender/non-transgender, or heterosexual/non-heterosexual.

### 3.2. Sex Differences in Predictors and Risk Factors for E-Cigarette Use

#### 3.2.1. Sex Differences in E-Cigarette Use and Initiation

Overall, studies considering the sex variable in e-cigarette initiation reported higher incidence for males than females [38,42,69,73], although others suggested no differences [18,25,47,51,58]. Stanton et al. recently suggested that the male sex is as a significant predictor for e-cigarette “Prior Initiation” and “Initiation with Progression” but not a significant predictor of cigarette trajectories [60]. In this regard, a study using a dependence scale reported no differences in dependence, but found that males and e-liquid smokers were more likely to quit e-cigarette smoking than females [111].

When asked about reasons to initiate, studies found that females consider e-cigarettes “less harmful than others” [43] and are motivated by being able to vape in areas where cigarette smoking is prohibited [22]. For males, e-cigarettes are considered “less harmful than cigarettes” and thus chosen as an alternative to smoking [43]. Males also view secondhand e-cigarette as harmless at higher rates than females [65]. Females, on the other hand, are more likely to perceive greater harm from e-cigarette use and find them as addictive as combustible cigarettes [49,65,119].

Initiation of e-cigarette use in females is mostly associated with curiosity [125]. However, females who never smoked are less likely to ever become users of e-cigarettes [31]. In general, girls tend to depend on their peers to access e-cigarettes as opposed to boys who obtain them independently [20]. For prior users, interestingly, both female smokers and those not identifying as heterosexual/straight are more likely to add e-cigarette use to their smoking habits than males [31].

In terms of e-cigarette and marketing, there are currently no sex differences reported in the generation of encouraging or discouraging message themes for the prevention of e-cigarette use [62,71]. Preventative strategies for e-cigarette initiation, such as parental monitoring and raising the minimum age for tobacco purchase, have been effective to reduce tobacco use, including e-cigarette use in both sexes [128,133]. While there was a previously evident sex difference with males using e-cigarette more than females, this disappeared by 2017, with social media messages displaying titles such as “How women are redefining the vape culture” and messages in support of female vaping (e.g., e-puffer 2020; Women Triangle 2019)”, which resulted in increases in vaping rates for females [5,129].

#### 3.2.2. Sex Differences in E-Cigarette Use for Smoking Cessation and Polysubstance Use

In general, studies on this topic reported that while women are more frequent users of e-cigarettes for smoking cessation purposes, e-cigarettes are more effective for actual cessation in men [34,43]. Males also report higher dependence motives, with cravings positively associated with unsuccessful quitting attempts, while females display weaker associations [72]. Regarding effectiveness as a smoking cessation tool, one study showed that gender does not influence e-cigarette adoption as a means to quit smoking [33]; a separate systematic review reported gender differences and found that men currently using e-cigarettes have a higher probability of having recently quit than women [70]. To explain this phenomenon, the possibility that men receive more nicotine from e-cigarettes than women was discussed as a factor contributing to e-cigarettes’ differential usefulness for cessation [70]. In this regard, males are more likely to report using e-liquid nicotine with concentrations above 20 mg/mL and a device capacity greater than 2 mL, whereas women use smaller devices [19]. Young female vapers tend to engage in low-power, high-nicotine-concentration patterns as opposed to men who choose high-power/lower-nicotine patterns (sub-ohming, cloud chasing) [31]. Males are also more satisfied using e-cigarettes as an aid for smoking cessation than women because they experience less cravings and withdrawal symptoms in comparison to conventional smoking [125].

Regarding characteristics of device choice, males more frequently use box-shaped tanks, whereas females prefer pen-style devices [43,125]. In this regard, users of open systems (rechargeable, do not use cartridges, and refillable) are mostly likely to be male as opposed to being users of closed systems (not rechargeable, or rechargeable and used cartridges) [34]. Females tend to prefer devices that resemble conventional cigarettes and pod-type/disposable devices, whereas males prefer tank- and mod-rechargeable types [67,68]. One study assessing pod use initiation also reported that males had significantly greater odds of future pod use than females after just 18 months of use [59]. Males buying pods also have higher odds of frequent e-cigarette use than females and those who purchased by the pack or box [44].

With regard to polysubstance/poly-product use, two reports suggested that young adult females have higher susceptibility to e-cigarette use than males when using other substances [35,51]. Being male is associated with higher rates of reuptake for prior users in all tobacco products, including e-cigarettes [39], and male e-cigarette users are more likely to become smokers than females [26,30,48]. Interestingly, a study recognized that young adult female waterpipe smokers are 12 times more likely to use e-cigarettes than young adult males [35]. Baseline alcohol use only predicts e-cigarette use among females, with both peer and parental e-cigarette smoking having a stronger effect on females than males [21,43,52].

#### 3.2.3. Sex Differences in E-Cigarette Flavor Use, Device Choice, and Preference

Flavors are strong drivers for e-cigarette use, with young adults who use tobacco-free nicotine e-cigarettes reporting the use of fruit, menthol, mint, and beverage flavors [32,40]. Moreover, the frequency of e-cigarette use and the use of nicotine e-liquid among young children has dramatically increased in the recent decade [22,43], putting children at a high risk of nicotine poisoning when fruit- or candy-flavored e-cigarette refills are available in their homes [37]. In general, studies addressing the sex variable show that flavors influence the choice of e-cigarettes in both males and females. Other studies show that while females prefer to vape fruit flavors [57], tobacco flavor is more popular among males [23]. Notably, males are more likely to be seen purchasing e-liquids at vaping shops than female customers [118].

A laboratory-controlled e-cigarette administration study in young adults aged 18–35 years found that females prefer menthol and fruit flavors to tobacco flavor [37]. Interestingly, this study also found that males find fruit flavors more appealing than tobacco or menthol [37]. Additionally, a study using custom e-cigarette flavors found that “cool” and “sweet” as well as “smooth” flavors have greater appeal in males when compared to females [66]. However, “bitter” and “harsh” flavors displayed strong negative associations for both males and female participants, with more robust negative associations in females [43]. Overall, males who experience pleasant e-cigarette sensory attributes may be more likely to find e-cigarettes appealing; however, sensory attributes are less likely to influence appeal in females [27]. Despite this, women report stronger sensory e-cigarette expectancies related to taste/smell and pleasure than men [27].

#### 3.2.4. Impact of the COVID-19 Pandemic on E-Cigarette Use

The COVID-19 pandemic played a significant role in e-cigarette and other substance initiation as a coping behavior [122]. Significant differences in the prevalence of using cannabis, alcohol, and vaping to cope with the effects of the pandemic were reported in one study, in which a larger proportion of females reported engaging in substance-specific behaviors than males [122]. However, others showed an increase in depressive symptoms but a decrease in e-cigarette use during the COVID-19 pandemic—a response that is thought to be the result of isolation [120]. Interestingly, another study found that both males and females had the same number of vaping episodes per day, although females took more puffs per vaping episode [50]. In general, females appeared to be more vulnerable to substance use relative to males during the COVID-19 pandemic [50].

#### 3.2.5. Other Sex-Specific Factors Influencing E-Cigarette Use

One reported predictor of e-cigarette use is the effect of vaping on weight. A study conducted in China found that obesity increased the odds of e-cigarette use [45]. When comparing males and females, adolescent girls who were trying to lose weight were at a higher risk for frequent vaping, while males showed no significant association [66].

Another factor affecting vaping choice is bullying in the form of school bullying and cyber bullying. One study found that female students who were bullied had higher odds of vaping than those who did not experience bullying [55], while others found that all bullying victims had statistically significantly higher odds of all measures of e-cigarette smoking [41]. For boys, this was only found in those bullied daily or almost daily when compared to boys not bullied [41].

Parental smoking has also been implicated in e-cigarette use. A survey-based study found that the male sex, peer, and parental use of e-cigarettes, as well as being unaware of the risks related with their use were independently associated with the use of e-cigarettes [54]. Unemployment status was another factor that was found to be associated with sex-differences in e-cigarette use during the pandemic, where no longer being employed was associated with vaping in females but not males [64]. In terms of adverse childhood experiences (ACE), a study in young adults found that cumulative ACE had stronger associations to current e-cigarette use among males but not females [116]. Finally, pain severity has also been associated with sex differences in e-cigarette health literacy, with women seeking information about e-cigarettes at low levels of pain, with no differences at high levels of pain [36].

### 3.3. Sex Differences in General Effects of E-Cigarette Use in Organ Systems

As mentioned earlier, while e-cigarettes were initially designed to mitigate the damaging effects caused by smoking, several studies have reported that the chemicals present in e-cigarette smoke and fluids are also dangerous to human health. These chemicals include, but are not limited to, acetaldehyde, acrolein, formaldehyde, and other aldehydes [134]. Upon heating, the vegetable glycerin present in e-liquids forms acrolein, whereas the propylene glycol forms acetaldehyde and formaldehyde, which are retained in the respiratory tract at dramatically high rates [135]. Also, one of the harmful constituents is vitamin E acetate, which has been found in the bronchoalveolar lavage fluid of patients with EVALI (e-cigarette or vaping product use associated lung injury), a condition found more prevalent in young males (67%) than females [136,137,138]. While much remains to be determined about the lasting health consequences of acute and chronic exposures to these chemicals, the literature presented below in clinical and preclinical/experimental studies not only shows that there are negative health effects in various organ systems, but also that there are sex differences in the susceptibility to these effects and associated toxicity.

Overall, the most prominent effects have been reported on respiratory, immune, and cardiovascular systems [74]. Interestingly, some of these effects are exacerbated with the dual use of e-cigarettes and combustible cigarettes [75]. In a crossover study of healthy subjects matched for age and sex, Carnevale et al. showed that e-cigarettes have less harmful vascular effects compared to combustible cigarettes based on oxidative stress and endothelial function changes [77]. Additionally, timing of use also influences these effects. For example, short-term vaping has been associated with elevated endothelial progenitor cells and microvesicles [76], whereas long-term vaping can predispose users to premature atherosclerosis and alter innate and adaptive immunity markers, pro-inflammatory cytokine secretion, and reactive oxidative species levels [78].

#### 3.3.1. Clinical Studies Assessing Organ System Effects in Males and Females

##### Cardiopulmonary Effects

A study conducted to assess the effect of cigarette smoking and vaping on the development of hypertension found that exclusive cigarette smoking, but not exclusive e-cigarette or dual e-cigarette + cigarette use, was associated with an increased risk of incident hypertension among females. However, neither e-cigarette nor cigarette use, when used exclusively or in tandem as part of a dual-use pattern, were associated with subsequent hypertension in males [79].

Regarding pulmonary disease, e-cigarette use has been found to cause significant obstructive lung function impairment in individuals who reported ever using an e-cigarette when compared to those who did not [81,82]. Associations of vaping with asthma also displayed sex-dependent effects. Recent data suggest that there are sex-dependent effects of e-cigarette use on the frequency of asthma symptoms, which are likely caused by hormonal differences, immune responses, and the susceptibility and toxicity to chemicals that are present in e-cigarette fluids and devices. While male e-cigarette users are found to be more likely to have more frequent asthma symptoms compared to non-e-cigarette users, this finding is not evident in female e-cigarette users [83]. This is potentially associated with sex differences in the susceptibility and toxicity to chemicals present in e-cigarette fluids and devices, as well as vaping habits.

##### Central Nervous System Effects

In terms of the central nervous system, a few studies have been conducted assessing sex differences, with special attention being paid to the effects of nicotine on the male and female brain [85]. Overall, the magnitude of brain nicotine accumulation after e-cigarette use is higher in women than in men [84]. In addition, nicotine poisoning has been reported because of e-cigarette use, with most cases occurring in males and symptoms including nausea, vomiting, and dizziness [86]. Other vaping-related injuries, with more reports in males and young individuals, comprise burns, asphyxia or poisoning, irritation or an allergic reaction (10.3%), and fractures caused by exploding batteries [132].

#### 3.3.2. Preclinical Studies Assessing Organ System Effects in Males and Females

Given the toxicity associated with chemical components present in e-cigarettes, several preclinical studies using animal models and cell exposure systems have evaluated the effects of vaping exposure on various organ systems. However, only a few have evaluated sex dependent effects. Table 2 shows a summary of the main findings from studies addressing effects in pulmonary, cardiovascular, and nervous systems.

Regarding cardiopulmonary effects, while males display more prominent effects than females for inflammatory markers [96,97] and cardiovascular responses [99], females show changes in airway resistance [98], indicating potential sex-specific physiological mechanisms. In this regard, alveolar type II epithelial cells exposed to e-cigarette vapor experience increased cell death, and trigger macrophage recruitment in the airways [102]. E-cigarette vapor exposure in mice also results in airway inflammation and an impaired immune response to bacteria and viruses, including defective bacterial phagocytosis [101]. To evaluate female-specific effects, a study using intact and ovariectomized female mice exposed to nicotine for 3 months revealed no differences in serum cotinine levels and no structural or functional cardiopulmonary dysfunction changes [106].

In almost all cases, particularly in central nervous system responses, the presence of nicotine in vaping fluids exacerbated the effects [94,95,103,104,105,108]. Not surprisingly, sex-dependent effects were found, whereby female adolescent rats displayed lower anxiolytic effects and were more sensitive to the rewarding effect of nicotine than their male counterparts [93,139,140], although others did not find sex differences [107]. Interestingly, the gut microbiome was found to mediate sex-dependent effects related to nicotine metabolism in chronic e-cigarette exposure, and nicotine exposure was found to decrease alpha diversity in females and beta diversity in both sexes [92]. A history of vaping was also associated with stroke, with higher odds for females [109].

Other substances that are added to e-cigarettes include THC, and its effects also appear to be sex- and dose-dependent [90,91]. Females exposed to THC exhibit higher concentrations of active metabolites in blood at lower concentrations as compared to males and therefore develop anxiety behaviors such as avoidance of open spaces and locomotor inhibition at lower doses [90]. Similarly, a study using rats found an overall increase in alcohol intake in adulthood in female rats compared to male rats when exposed to vaporized nicotine during adolescence, indicating a potential sex-dependent physiological effect mediating polysubstance use [89].

### 3.4. Sex Differences in Behavioral Effects and Gender-Specific Effects of E-Cigarette Use

#### 3.4.1. Sex-Dependent Behavioral Effects Associated with E-Cigarette Use

Works conducted in preclinical models and population studies have identified associations of exposures to vaping products with a variety of behavioral effects, including attitude, eating disorders, preference for sports, depression, and sexual behaviors. Some of these studies examined the sex (biological) variable while others focused on self-identified genders. Not surprisingly, some of these effects are influenced by nicotine content [114]. Regarding attitudes, Aghar et al. identified an association with a positive attitude and male sex with e-cigarette use, where self-reported male users were six times more likely to have a positive attitude compared to e-cigarette users, whereas females were four times less likely to harbor a positive attitude compared to males [130]. Regarding worry and e-cigarette cognition, male e-cigarette users who experience worry may be especially vulnerable to greater maladaptive e-cigarette perceptions and cognition, including higher perceived benefits and lower negative consequences [117]. Another interesting finding in a study focusing on gender differences in e-cigarette users and sports participation was that users identifying as men were more likely to participate in sports than non-users [110]. However, these associations were not significant in females. Additionally, female e-cigarette users were found to be more sedentary compared to non-e-cigarette users [110].

In terms of racial disparities, in 2015, the greatest disparity was observed among White males, for whom the prevalence was 26.7%, and Black males, who had the lowest prevalence at 10.3%, a disparity of 16.4. Moreover, in 2019, the greatest disparity was seen between White females, at 14.4%, and Black females, at 4.9%. Correspondingly, the association of discrimination among Black and White U.S. adults aged 18–28 and e-cigarette use found that discrimination was directly associated with e-cigarette use only in the overall sample and indirectly associated through psychological distress among all race–sex groups except White males [121].

In regard to the association of vaping with eating disorders and depression, studies reported that female adolescents using e-cigarettes have greater chances of experiencing depression, suicidal ideation, and making a suicide plan than their male counterparts [113,124], while others found no differences [61]. When comparing dual users to non-users, females had higher odds of performing a suicide attempt than males.

#### 3.4.2. E-Cigarette Use in Sexual Minorities

Recently, multiple studies reported evidence of an increased prevalence of e-cigarette smoking among sexual minorities [39,53,112]. Importantly, these studies considered sex along with sexual orientation, but did not control for confounders such as stressors (e.g., victimization, rejection, concealment, discrimination, prejudice and parental disapproval, and/or depression), which could be associated with vaping among sexual minority groups.

A study on age-based patterns of five substance use behaviors across groups of adolescents defined by sexual orientation and gender identity showed that sexual minority girls reported the highest rates of substance use across all ages, followed by sexual minority boys [46]. This study showed that transgender youth were also more likely to report e-cigarette use [46]. While this study did not control for stressors, it mentioned that unique minority stressors, including victimization, rejection, and concealment, may spark substance use as a maladaptive coping strategy. Overall, sexual minority youth (SMY) has been found to have higher prevalence of e-cigarette use compared to heterosexuals across various studies [131], which is potentially related to disproportionate marketing of tobacco products to SMY, especially Black communities. Within this group, Non-Hispanic Black SMY were found to have higher prevalence of e-cigarette use compared to Black heterosexuals, which is potentially linked to the fact that SMY experience more discrimination that increases mental health risks [127]. This study highlighted the importance of exploring how other stressors (e.g., discrimination and prejudice, parental disapproval) among sexual minority youth could disproportionately impact current prevention efforts in these populations.

## 4. Discussion

The use of e-cigarettes is rapidly gaining popularity through social media platforms and advertisements targeting adults as well as younger individuals. The toxicity and negative health and behavioral effects associated with their use has been studied ever since devices were launched on the market [1]. However, the specific effects in male and female individuals have not been explored in detail, despite reports of differential toxicity and similarities of toxicants present in e-cigarettes with environmental pollutants that are known to trigger sex-specific effects [136,141]. In this study, we reviewed the available literature considering male and female subjects and by comparing various effects in males vs. females. After performing a systematic selection based on specific criteria (Table 1), we identified 115 studies that considered the sex variable in reporting the effects of various types of e-cigarette exposures. Overall, our review identified multiple sex-specific factors in organ system toxicities and exacerbation of diseases from preclinical and clinical studies, but also multiple gender differences in factors related to the initiation and frequency of use, as well as behavioral effects.

Our literature review identified various factors previously associated with e-cigarette use, including its use for smoking cessation, pleasure, and experimentation, as well as weight loss [43,66,70]. Others recognized e-cigarette use as a coping tool for bullying, depression, and psychological distress [41,55,124]. In addition, several preclinical and clinical studies reported sex-specific effects on cardiopulmonary, nervous, and immune systems (Table 2), as well as multiple behavioral outcomes. Overall, while males appear to be more prone to engage in e-cigarette use in several studies, we identified several factors that appear to encourage the use in females more than in males. These include fruit flavors, devices with low-power high-nicotine concentrations, pen- and pod-style rechargeable devices, polysubstance use, and even unemployment during the COVID-19 pandemic [122].

Furthermore, the health effects associated with e-cigarette use also displayed sex-specific characteristics in a variety of animal models and clinical studies. In general, females develop greater airway resistance following e-cigarette use. Odds of COPD and self-reported COPD as well as symptoms of asthma are greater in females as compared to males. The magnitude of nicotine accumulation in the brain is also greater in females than in males who vape. Also, pregnant women who use e-cigarettes have a greater risk of a small-for-gestational-age birth; however, quitting decreases this risk to non-smoker levels. One recent study has also discovered an effect on the gut microbiome, opening doors for future research on additional mechanisms of the brain– and lung–gut axes.

Preclinical studies analyzing behavior found that nicotine in e-liquids is associated with lower anxiolytic effects and higher sensitivity to rewarding effects in females exposed to e-cigarette smoke. Another constituent, THC, forms its active metabolite at lower concentrations in females as compared to males; therefore, females are at risk of developing anxiety and locomotor inhibition at lower doses. Moreover, an increased use of e-cigarettes with other substances, such as alcohol, has also been studied in animal models, with female adult rats increasing alcohol intake if they have history of nicotine vapor exposure in adolescence.

Understanding sex differences in factors associated with e-cigarette use can serve as a guide in the formation of targeted interventions. This study highlights potential disparities on the vulnerability to initiation, toxicity, and negative health effects for both males and females. Better knowledge of these effects can inform regulatory agencies to promote prevention and monitoring practices. For example, regulations such as raising the minimum age to limit access and parental monitoring have been found to be effective prevention strategies and can help fight the e-cigarette epidemic, some of which have displayed better efficacy in females than males. Similarly, sex-specific factors such as pregnancy could be considered in future studies. In this regard, some studies have addressed the effects of e-cigarette use during pregnancy [142] and found that continued users of e-cigarettes during pregnancy were at a higher risk of small-for-gestational-age birth than non-users, whereas quitters of e-cigarettes had a similar risk than non-users [87]. Similar findings were observed in women who used e-cigarettes before pregnancy and continued e-cigarette use during pregnancy, where those who continued had a significantly increased risk of small-for-gestational-age birth than those who quit [88].

When conducting this literature review, we faced several limitations. First, many of the studies available use the terms sex and gender interchangeably, limiting our ability to evaluate biological vs. social/behavioral effects. Second, while studies identified and reported sex differences in outcomes, many of these were not initially designed with the goal of assessing sex or gender variables; therefore, they lacked statistical power. Third, experimental and preclinical studies were highly variable in terms of timing, concentration, type of e-liquid, type of device, age and strain of animals, type of cells, and measured outcomes. Fourth, studies addressing gender effects and other factors, such as bullying, focused on associations and did not address confounding factors with adequate statistical approaches, limiting their interpretation. While many studies reported similar results in the directionality of effects, the available literature was not optimal to perform meta-analyses on the sex and gender variables. In this regard, it is important to highlight that in the studies found, e-cigarette use was often only one of many potential determinants of health outcomes, and therefore reliable conclusions can only be drawn using multivariate analysis models, which in most cases were not conducted. Finally, cohort studies that focused on behaviors were conducted in selected populations, lacking sufficient diversity to assess effects such as race, ethnicity, and socioeconomic factors. Subsequently, we were able to find a few studies addressing factors and health effects in sexual minorities which identified multiple challenges faced by these populations that were related to e-cigarette use.

In summary, the available studies incorporating sex or gender in health consequences of e-cigarette use from the past five years collectively show that there are differences in risk factors, motivation, behaviors, and organ system effects between males and females. Many of these factors were influenced by interventions or social factors, while others appeared to be biological. Further investigations are required to determine the mechanisms underlying these differences, as well as the interactions of sex and gender factors in influencing the health consequences of using these rapidly expanding devices in both youth and adults.

## 5. Conclusions

To date, many studies have explored sex and gender variables as the modulators of the effects of e-cigarette use in physiological and behavioral outcomes. Collectively, the available literature supports the notion that both sex and gender influence the susceptibility to negative e-cigarette health effects as well as factors influencing its use in different populations. Therefore, future research needs to consider sex and gender variables when addressing e-cigarette toxicity and other health-related consequences in order to better inform preventative strategies and educate users.

## Figures and Tables

**Figure 1 ijerph-20-07079-f001:**
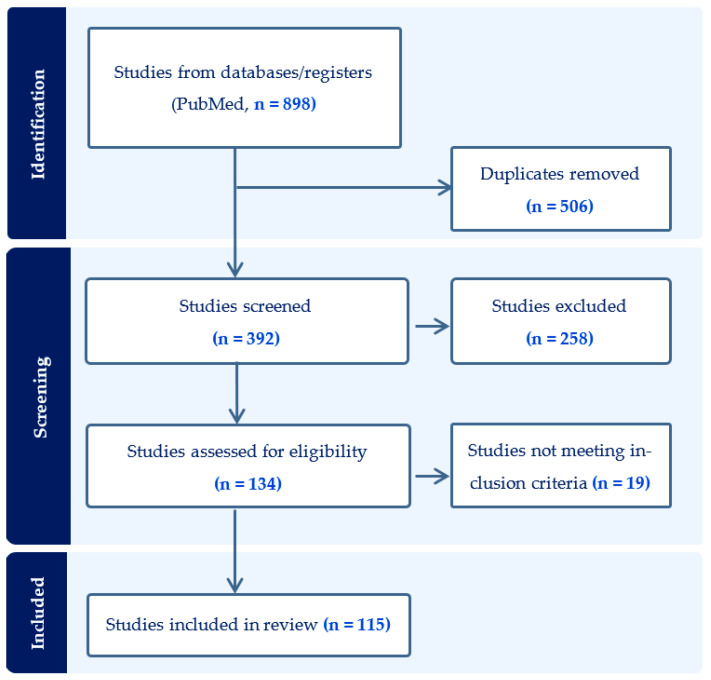
Search strategy and results.

**Table 1 ijerph-20-07079-t001:** Types of e-cigarette studies incorporating the sex variable.

Topic	# of Articles	Refs.
Sex differences in predictors, risk factors, and epidemiology.	56	[18,19,20,21,22,23,24,25,26,27,28,29,30,31,32,33,34,35,36,37,38,39,40,41,42,43,44,45,46,47,48,49,50,51,52,53,54,55,56,57,58,59,60,61,62,63,64,65,66,67,68,69,70,71,72,73]
Sex differences in organ systems effects (clinical and preclinical).	36	[74,75,76,77,78,79,80,81,82,83,84,85,86,87,88,89,90,91,92,93,94,95,96,97,98,99,100,101,102,103,104,105,106,107,108,109]
Sex and gender differences in behavioral effects.	23	[110,111,112,113,114,115,116,117,118,119,120,121,122,123,124,125,126,127,128,129,130,131,132]

**Table 2 ijerph-20-07079-t002:** Sex-dependent effects of e-cigarette exposures.

Study	Exposure/Model	Results
Pulmonary		
Wang et al., 2019 [96]	Acute (2 h/day) 3-day exposure to e-cigarette aerosols (Joytech eVIC VTC mini-ENDS) containing PG +/-nicotine (vs. air-exposed controls) in C57BL/6J mice (14–16 weeks old).	Males only: increased BALF IL-3, IL-4, IL-9, IL-12p70, IFNγ, GM-CSF, Eotaxin, and MIP-1β with PG+nicotine; increased CTNN1B with PG only. Females only: increased BALF neutrophil CD8a+ T-lymphocytes and IL-1β, and lung tissue ADRP with PG+nicotine; increased lung tissue ADRP and PPARγ with PG+/−nicotine. Both sexes: higher BALF MPO activity and lung tissue ADRP with PG only; lower MPO activity with PG+nicotine; higher lung tissue nAChRα3 and nAChRα7 in both PG+/−nicotine (effect more robust in females).
Lallai et al., 2021 [97]	PV/VG + nicotine vapor exposure (1 h/day, 5 days, 1 puff every 5 min) in a sealed chamber (adult C57BL6/J mice).	Males only: increased ACE2 mRNA expression and cell density in the lungs exposed to nicotine. Both sexes: increased cotinine levels, downregulated nAChR α5 subunit.
Naidu et al., 2021 [98]	A 30 min e-cigarette vapor exposure (+/−nicotine), 2 times/day for 21 days (adult BALB/c mice).	Females only: significant increase in AHR with nicotine exposure. Both sexes: vapor increased BALF MCP-1, IL-1β, and KC levels. Nicotine induced lung ACE-2 expression (higher in males).
Wang et al., 2020 [100]	Pregnant CD-1 mice exposed to e-cigarette aerosols (PV/VG +/- nicotine), 3 h/day, 5 days/week, 3 weeks. Adult (6-weeks old) offspring analyzed.	Females only: upregulated LEF-1, HDAC-1, and fibronectin in pups exposed to PG/VG. Males only: increased lung PPARγ, CNN1, ACTA2, and α-SMA; decreased e-cadherin in pups exposed to PG/VG+nicotine. Both sexes: increased PAI-1 and decreased MMP9 levels in pups exposed to PG/VG+nicotine.
Cardiovascular		
Carll et al., 2022 [99]	Performed 9 min puff sessions three times; 2 sessions with PG/VG and menthol (bluPLUS+ cartridges) in telemetered C56BL/6/mice (12–30 weeks old).	Males only: significant HR and HRV responses and solvent-induced bradyarrhythmias and bradycardia than females. More susceptible to bradypnea, cardiac depression, and mortality upon acute exposures to high acrolein, and had higher VPBs with exposure to menthol-containing aerosols. Both sexes: e-cigarette exposure increased the frequency of ventricular tachyarrhythmias.
Central nervous system		
Nguyen et al., 2020 [90]	PG or THC vapor exposure 2 times/day for 30 min using an e-cigarette system in Wistar adolescent and adult rats.	Males only: consumed more food after repeated adolescent THC and had significantly lower body weight during the second treatment week. Females only: developed rapid tolerance (adolescents); self-administered more fentanyl with repeated THC exposure. Both sexes: hypothermic after THC vapor inhalation; had persisting tolerance as adults.
Honeycutt et al., 2020 [94]	Vaporized nicotine exposure once daily for 5 days in C56BL/6 adult mice.	Females only: more sensitive to hypothermic effects. No sex differences in locomotor activity.

Abbreviations—α-SMA: alpha-smooth muscle actin; ADRP: adipocyte differentiation-related protein; AHR: airway hyperresponsiveness; BALF: bronchoalveolar lavage fluid; CNN1: calponin 1; CTNNB1: β-catenin; GM-CSF: granulocyte–macrophage colony-stimulating factor; HDAC-1: histone deacetylase 1; HR: heart rate; HRV: heart rate variability; IFNγ: interferon γ; IL-: interleukin; LEF-1: lymphoid enhancer-binding factor 1; MCP-1: monocyte chemoattractant protein-1; MPO: myeloperoxidase; nAChR: nicotinic acetylcholine receptor; PG: polyethylene glycol; PAI-1: plasminogen activator inhibitor-1; PND: postnatal day; VG: vegetable glycerin; VPBs: ventricular premature beats.

## Data Availability

Details of the literature search can be found at: https://app.covidence.org/reviews/352715, URL (accessed on 11 October 2023).

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
