# Peer review of "Sex Differences in E-Cigarette Use and Related Health Effects"

_ijerph, 2023, doi:10.3390/ijerph20227079_

Round 1

Reviewer 1 Report

Comments and Suggestions for Authors

This exceptionally important research aims to review the gender differences in the use of e-cigarettes and the health consequences of their use.

I would like to draw the attention of the authors to two problems, which they could consider to further refine their manuscript.

1.      Among the limitations of the research, they write: "Firstly, many of the studies available use the terms sex and gender interchangeably, limiting our ability to evaluate biological vs. social/behavioral effects." (lines 503-4) However, apart from studies where the sex of the participants is biologically confirmed, such as in pregnant women, the participants themselves determine their sex, which is considered to be gendered. Except in studies where the sex of the participants is biologically confirmed, such as in pregnant women, the participants themselves determine their sex, which is considered to be gendered. As the vast majority of the studies reviewed were surveys, where we can be sure that gender was identified through self-determination, it should be made clearer from the beginning of the study that the results presented are predominantly gender differences.

2.      In light of the results presented, the reader may wonder how well each study controlled for confounders. For example, the authors state that „Others recognized e-cigarette use as a coping tool for bullying, depression, and psychological distress.” (lines 470-1) At another place in the text, „Recently, multiple studies reported evidence of increased prevalence of e-cigarette smoking among sexual minorities.” (lines 426-7) However, there is no mention of whether the research on sex minorities controlled for bullying. It should be stressed that e-cigarette use is only one of the determinants of health outcomes, and therefore reliable conclusions can only be drawn using multivariate analysis models.

Author Response

Reviewer 1

This exceptionally important research aims to review the gender differences in the use of e-cigarettes and the health consequences of their use.

I would like to draw the attention of the authors to two problems, which they could consider to further refine their manuscript.

Among the limitations of the research, they write: "Firstly, many of the studies available use the terms sex and gender interchangeably, limiting our ability to evaluate biological vs. social/behavioral effects." (lines 503-4) However, apart from studies where the sex of the participants is biologically confirmed, such as in pregnant women, the participants themselves determine their sex, which is considered to be gendered. Except in studies where the sex of the participants is biologically confirmed, such as in pregnant women, the participants themselves determine their sex, which is considered to be gendered. As the vast majority of the studies reviewed were surveys, where we can be sure that gender was identified through self-determination, it should be made clearer from the beginning of the study that the results presented are predominantly gender differences.

Response: We thank the reviewer for his/her comments. We have edited the statement in lines 503-4 to clarify what our findings revealed about the sex and gender variables in assessing various outcomes. We have also provided additional clarification in the revised introduction and methods regarding consideration of these variables, including consideration of the sex (biological) variable in preclinical studies, and self-reporting of gender in behavioral and survey-based studies. Concerning studies in pregnancy, we have removed this section from the results and added a sentence about pregnancy in the discussion for consideration in future research.

In light of the results presented, the reader may wonder how well each study controlled for confounders. For example, the authors state that „Others recognized e-cigarette use as a coping tool for bullying, depression, and psychological distress.” (lines 470-1) At another place in the text, „Recently, multiple studies reported evidence of increased prevalence of e-cigarette smoking among sexual minorities.” (lines 426-7) However, there is no mention of whether the research on sex minorities controlled for bullying. It should be stressed that e-cigarette use is only one of the determinants of health outcomes, and therefore reliable conclusions can only be drawn using multivariate analysis models.

Response: We thank the reviewer for his/her recommendations. We have edited the discussion and limitations section to clarify the consideration of bullying as a confounder in studies, as well as the need for more comprehensive statistical analyses to assess health outcomes. We hope that the revisions to the manuscript have improved its quality.

Reviewer 2 Report

Comments and Suggestions for Authors

Sex differences in e-cigarette use and health effects

Thank you for the opportunity to review this manuscript, which reviewed literature regarding sex differences in e-cigarette use and the health effects of such use. While this is an interesting review with the potential for significant public health impact, there were several limitations that limited my enthusiasm. I have outlined my concerns below in the hope that the authors find these comments useful.

Title:

1.       I recommend slightly editing the title to be more specific – “Sex differences in e-cigarette use and related health effects”

Abstract:

1.       “The main differences reported were related to reasons for initiation including smoking history, types of devices and flavoring, polysubstance use, physiological responses to nicotine and toxicants in e-liquids, exacerbation of lung disease, and behavioral factors like anxiety, depression, sexuality, and bullying, and sexuality.” – Sexuality is listed twice

2.       Gender is mentioned in the conclusion of the abstract, but is not mentioned earlier. Given the purpose of the review, it is important to be extremely clear regarding sex versus gender to avoid confusion. I recommend either incorporating gender earlier in the abstract or removing it from the conclusion.

Introduction:

1.       “These e-cigarettes’ constituents play a role in affecting health, as indicated by multiple studies conducted in populations, animal models, and cells.” – I recommend specifying human populations here.

2.       I recommend editing the first sentence of paragraph 2, as vaping has gained popularity among youth over the past decade. The way it is currently worded makes I sound like it is just now gaining popularity. Moreover, in recent years, we are seeing slight declines in youth vaping, which may be important to incorporate into the introduction.

3.       “Not only do these devices deliver a greater concentration of nicotine, but also many currently available vape products contain tetrahydrocannabinol (THC) and other compounds that are unknown to the user.” I recommend being more specific here – a greater concentration of nicotine relative to what?

4.       I recommend providing a slightly more balanced view of the health effects of e-cigarettes by briefly mentioning the benefits if used as a cigarette cessation aid among adults.

5.       The third sentence in paragraph 3 is a bit hard to follow. I recommend breaking this up and expanding on these points, as sex differences are the topic of this paper, but are only briefly mentioned in the introduction.

Results:

1.       In the results, it is mentioned that studies were excluded due to setting, but setting does not seem to be mentioned as an inclusion/exclusion criteria. Please clarify.

2.       “In the latter category, several studies used the terms sex (biological variable) and gender (social construct) interchangeably, without including a definition of these terms, so articles using both terms were included.” This is a bit concerning, as these are very different constructs and should not be used interchangeably. If the articles did not provide a definition, could the authors provide a bit more information – for instance, were these articles referring to participants as men/women or male/female? Did they specify cisgender/transgender? This could be a confounding factor of the review that needs to be addressed.

3.       I recommend thoroughly editing the results. They are extremely long and hard to digest. As this is a review, it would be helpful to present overall findings across studies (eg, “Most studies suggest…”), rather than specifically mentioning findings separately study by study. This is a theme throughout the entire results section.

4.       It is mentioned in both sections 3.2.1 and 3.2.2 that males are more likely to escalate to cigarette use than females. Given that the results are so long, it is important that they are well organized and that information presented is consistent with the subheading.

5.       It is unclear why vaping during pregnancy is reviewed, as there are no sex differences presented. I recommend removing this from the results.

Discussion:

1.       In the first paragraph of the discussion, it is mentioned that the review focused on sex or gender differences in e-cigarettes. However, this is not mentioned as the focus throughout the introduction. If this is going to be framed as a review on sex and/or gender differences, there needs to be far more discussion of this in the introduction, including a definition of these constructs. I recommend that the authors remove gender, as their focus appears to be primarily on sex differences and their search criteria focused on sex, rather than gender.

2.       The information presented throughout the discussion needs to be cited.

3.       The discussion is largely a repetition of the results. There needs to be an explanation of findings and greater discussion of the implications of such findings.

Other:

1.       In several places, there is mention of “e-cigarettes use” or “e-cigarettes research.” This should be “e-cigarette use” or “e-cigarette research.”

Comments on the Quality of English Language

In several places, there is mention of “e-cigarettes use” or “e-cigarettes research.” This should be “e-cigarette use” or “e-cigarette research.”

Author Response

Reviewer 2

Thank you for the opportunity to review this manuscript, which reviewed literature regarding sex differences in e-cigarette use and the health effects of such use. While this is an interesting review with the potential for significant public health impact, there were several limitations that limited my enthusiasm. I have outlined my concerns below in the hope that the authors find these comments useful.

Title:

I recommend slightly editing the title to be more specific – “Sex differences in e-cigarette use and related health effects”

Response: We thank the reviewer for his/her recommendation. We agree that this is a better title and thus have changed it.

Abstract:

  1. “The main differences reported were related to reasons for initiation including smoking history, types of devices and flavoring, polysubstance use, physiological responses to nicotine and toxicants in e-liquids, exacerbation of lung disease, and behavioral factors like anxiety, depression, sexuality, and bullying, and sexuality.” – Sexuality is listed twice
  2. Gender is mentioned in the conclusion of the abstract, but is not mentioned earlier. Given the purpose of the review, it is important to be extremely clear regarding sex versus gender to avoid confusion. I recommend either incorporating gender earlier in the abstract or removing it from the conclusion.

Response to 1&2: We appreciate the reviewer for finding out the error. We have corrected the abstract and edited its content for clarity. We have also provided additional clarification regarding the sex and gender variables throughout the manuscript.

Introduction:

  1. “These e-cigarettes’ constituents play a role in affecting health, as indicated by multiple studies conducted in populations, animal models, and cells.” – I recommend specifying human populations here.
  2. I recommend editing the first sentence of paragraph 2, as vaping has gained popularity among youth over the past decade. The way it is currently worded makes I sound like it is just now gaining popularity. Moreover, in recent years, we are seeing slight declines in youth vaping, which may be important to incorporate into the introduction.

Response to 1&2: We thank the reviewer for his/her recommendations. We have specified human populations and edited the introduction to reflect the most recently published statistics regarding youth e-cigarette use.

  1. “Not only do these devices deliver a greater concentration of nicotine, but also many currently available vape products contain tetrahydrocannabinol (THC) and other compounds that are unknown to the user.” I recommend being more specific here – a greater concentration of nicotine relative to what?
  2. I recommend providing a slightly more balanced view of the health effects of e-cigarettes by briefly mentioning the benefits if used as a cigarette cessation aid among adults.
  3. The third sentence in paragraph 3 is a bit hard to follow. I recommend breaking this up and expanding on these points, as sex differences are the topic of this paper, but are only briefly mentioned in the introduction.

Response to 3,4&5: We thank the reviewer for all these recommendations. We have thoroughly revised the introduction to specify comparisons regarding nicotine concentration, discuss the health effects and benefits of e-cigarette use as a smoking cessation tool, and break up the last section to clarify sex differences in inhalation toxicology and how they relate to this paper. 

Results:

  1. In the results, it is mentioned that studies were excluded due to setting, but setting does not seem to be mentioned as an inclusion/exclusion criteria. Please clarify.

Response: We appreciate the reviewer for identifying the issue. The mention of “setting” was incorrect and has been removed in the revised version.

  1. “In the latter category, several studies used the terms sex (biological variable) and gender (social construct) interchangeably, without including a definition of these terms, so articles using both terms were included.” This is a bit concerning, as these are very different constructs and should not be used interchangeably. If the articles did not provide a definition, could the authors provide a bit more information – for instance, were these articles referring to participants as men/women or male/female? Did they specify cisgender/transgender? This could be a confounding factor of the review that needs to be addressed.

Response: To address the reviewer’s comment (and another reviewer’s suggestion) the revised manuscript provides additional clarification about the definition and use of the sex and gender variables, including consideration of the sex (biological) variable in preclinical studies, and self-reporting of gender in behavioral and survey-based studies.

  1. I recommend thoroughly editing the results. They are extremely long and hard to digest. As this is a review, it would be helpful to present overall findings across studies (eg, “Most studies suggest…”), rather than specifically mentioning findings separately study by study. This is a theme throughout the entire results section.
  2. It is mentioned in both sections 3.2.1 and 3.2.2 that males are more likely to escalate to cigarette use than females. Given that the results are so long, it is important that they are well organized and that information presented is consistent with the subheading.

Response to 3&4: We thank the reviewer for his/her suggestions to improve the results section. We have now edited the manuscript to summarize overall findings when possible, and we have incorporated introductory sentences to each results subsection to make our study easier to read. We hope that these revisions have improved the manuscript’s quality.

  1. It is unclear why vaping during pregnancy is reviewed, as there are no sex differences presented. I recommend removing this from the results.

Response: To address the reviewer’s comment (and another reviewer’s suggestion) the revised manuscript does not include studies in pregnancy. We have instead added a sentence about this topic in the discussion for consideration in future research.

Discussion:

  1. In the first paragraph of the discussion, it is mentioned that the review focused on sex or gender differences in e-cigarettes. However, this is not mentioned as the focus throughout the introduction. If this is going to be framed as a review on sex and/or gender differences, there needs to be far more discussion of this in the introduction, including a definition of these constructs. I recommend that the authors remove gender, as their focus appears to be primarily on sex differences and their search criteria focused on sex, rather than gender.

Response: We thank the reviewer for the suggestion. We have clarified the consideration of each variable in the revised discussion. While our intent was to summarize sex-specific findings, many of the studies reported gender-related behavioral factors that we considered important to include.

  1. The information presented throughout the discussion needs to be cited.

Response: We apologize for the oversight. We have added additional references in the revised discussion.

  1. The discussion is largely a repetition of the results. There needs to be an explanation of findings and greater discussion of the implications of such findings.

Response: We thank the reviewer for his/her suggestions to improve the discussion section. We have now expanded this section to incorporate additional comments regarding the studies’ findings and their implications.  

Other:

  1. In several places, there is mention of “e-cigarettes use” or “e-cigarettes research.” This should be “e-cigarette use” or “e-cigarette research.”

Response: We appreciate the reviewer’s recommendation. We have edited the wording accordingly throughout the whole manuscript.

Round 2

Reviewer 2 Report

Comments and Suggestions for Authors

Several of my concerns were addressed. However, the results remain extremely long (9 pages). It seems as though the authors added sentences summarizing the results but did not present the results more succinctly. As I mentioned previously, the results are difficult to follow given that findings are presented study by study rather than discussing common themes. There remain many typos throughout the manuscript as well. 

Comments on the Quality of English Language

There are several typos and grammatical errors throughout the manuscript.

Author Response

Reviewer comment: Several of my concerns were addressed. However, the results remain extremely long (9 pages). It seems as though the authors added sentences summarizing the results but did not present the results more succinctly. As I mentioned previously, the results are difficult to follow given that findings are presented study by study rather than discussing common themes. There remain many typos throughout the manuscript as well. 

Response: We thank the reviewer for the additional suggestions. We have shortened the results to 7 pages and reorganized the information accordingly. Spelling and grammar were checked and typos were corrected.